# BAYESIAN ACTIVE LEARNING BY DISTRIBUTION DIS-AGREEMENT

## ABSTRACT

Active Learning (AL) for regression has been systematically under-researched due to the increased difficulty of measuring uncertainty in regression models. Since normalizing flows offer a full predictive distribution instead of a point forecast, they facilitate direct usage of known heuristics for AL like Entropy or Least-Confident sampling. However, we show that most of these heuristics do not work well for normalizing flows in pool-based AL and we need more sophisticated algorithms to distinguish between aleatoric and epistemic uncertainty. In this work we propose BALSA, an adaptation of the BALD algorithm, tailored for regression with normalizing flows. With this work we extend current research on uncertainty quantification with normalizing flows (Berry and Meger, 2023b;a) to real world data and pool-based AL with multiple acquisition functions and query sizes. We report SOTA results for BALSA across 4 different datasets and 2 different architectures.

## 1 INTRODUCTION

The ever growing need for data for machine learning science and applications has fueled a long history of Active Learning (AL) research, as it is able to reduce the amount of annotations necessary to train strong models. However, most research was done for classification problems, as it is generally easier to derive uncertainty quantification (UC) from classification output without changing the model or training procedure. This feat is a lot less common for regression models, with few historic exceptions like Gaussian Processes. This leads to regression problems being under-researched in AL literature. In this paper, we are focusing specifically on the area of regression and recent models with uncertainty quantification (UC) in the architecture. Recently, two main approaches of UC for regression problems have been researched: Firstly, Gaussian neural networks (GNN) (Flunkert et al., 2017; Madhusudhanan et al., 2024), which use a neural network to parametrize $\mu$ and $\sigma$ parameters and build a Gaussian predictive distribution and secondly, Normalizing Flows (Papamakarios et al., 2017; Durkan et al., 2019), which are parametrizing a free-form predictive distribution with invertible transformations to be able to model more complex target distributions. Their predictive distributions allow these models to not only be trained via Negative Log Likelihood (NLL), but also to draw samples from the predictive distribution as well as to compute the log likelihood of any given point $y$. Recent works (Berry and Meger, 2023b;a) have investigated the potential of uncertainty quantification with normalizing flows by experimenting on synthetic experiments with a known ground-truth uncertainty.

Intuitively, a predictive distribution should inertly allow for a good uncertainty quantification (e.g. wide Gaussians signal high uncertainty). However, we show empirically that 2 out of 3 well-known heuristics for UC, (standard deviation, least confidence and Shannon entropy) significantly underperform when used as acquisition functions for AL. We argue that this is due to the inability of these heuristics to distinguish between epistemic uncertainty (model underfitting) and aleatoric uncertainty (data noise), out of which AL can only reduce the former. To circumvent this problem, (Berry and Meger, 2023b;a) have proposed ensembles of normalizing flows and studied their approximations via Monte-Carlo (MC) dropout. Even though (Berry and Meger, 2023b;a) have demonstrated good uncertainty quantification, their experiments are conducted on simplified AL use cases with synthetic data. They have not benchmarked their ideas against other SOTA AL algorithms or used real-world datasets. In this work we propose a total of 4 different extensions of the BALD algorithm for AL, which relies on MC dropout to separate the two types of

uncertainty. We adapt BALD's methodology for models with predictive distributions, leveraging the distributions directly instead of relying on aggregation methods like Shannon entropy or standard deviation. Additionally, we extend well-known heuristic baselines for AL to models with predictive distributions. We report results for GNNs and Normalizing Flows on 4 different datasets and 3 different query sizes.

With a recent upswing in the area of comparability and benchmarking (Rauch et al., 2023; Ji et al., 2023; Lüth et al., 2024; Werner et al., 2024), we now have reliable evaluation protocols, which help us to provide an experimental suite that is reproducible and comparable.

Our code is available under: `https://anonymous.4open.science/r/Bayesian-Active-Learning-By-Distribution-Disagreement-8682/`

### CONTRIBUTIONS

- Three heuristic AL baselines for models with predictive distributions and three adaptations to the BALD algorithm for this use case, creating a comprehensive benchmark for AL with models with predictive distributions

- Two novel extensions of the BALD algorithm, which leverage the predictive distributions directly instead of relying on aggregation methods, which we call $\underline{B}$ayesian $\underline{A}$ctive $\underline{L}$earning by Di$\underline{S}$tribution Dis$\underline{A}$greement (BALSA)

- Extensive comparison of different versions of BALD and BALSA on 4 different regression datasets and 2 model architectures

## 2 PROBLEM DESCRIPTION

We are experimenting on pool-based AL with regression models. Mathematically we have the following:

Given a dataset $\mathcal{D}_{\text{train}} := (x_i, y_i) \quad i \in \{1, \ldots, N\}$ with $x \in \mathcal{X}, y \in \mathcal{Y}$ (similarly we have $\mathcal{D}_{\text{val}}$ and $\mathcal{D}_{\text{test}}$) we randomly sample an initial labeled pool $L^{(0)} \sim \mathcal{D}_{\text{train}}$ that we call the seed set. We suppress the labels from the remaining samples to form the initial unlabeled pool $U^{(0)} = \mathcal{D}_{\text{train}}/L^{(0)}$. We define an acquisition function to be a function that selects a batch of samples of size $\tau$ from the unlabeled pool $a(U^{(i)}) := \{x_b^{(i)}\} \in U^{(i)} \quad b := [0, \ldots, \tau]$. We then recover the corresponding labels $y_b^{(i)}$ for these samples and add them to the labeled pool $L^{(i+1)} := L^{(i)} \cup \{(x_b^{(i)}, y_b^{(i)})\}$ and $U^{(i+1)} := U^{(i)}/\{x_b^{(i)}\} \quad b := [0, \ldots, \tau]$. The acquisition function is applied until a budget $B$ is exhausted.

We measure the performance of a model $\hat{y} : \mathcal{X} \to \mathcal{Y}$ on the held out test set $\mathcal{D}_{\text{test}}$ after each acquisition round by fitting the model $\hat{y}^{(i)}$ on $L^{(i)}$ and measuring the Negative Log Likelihood (NLL)

## 3 BACKGROUND

### UNCERTAINTY QUANTIFICATION IN REGRESSION MODELS

Uncertainty quantification (UC) in regression models can broadly be archived by two approaches: (i) The architecture of the regression model is set up to produce an UC itself, or (ii) the training or inference of a model is subjected to an additional procedure to generate UCs.

Examples of (i) are Gaussian Processes and density-based models, which use an encoder to produce the parameters of a predictive distribution. The most common example is a Gaussian neural network (GNN), where the encoder produces the mean and variance parameters which create a Gaussian predictive distribution. Recently, Normalizing flows (NF) have been proposed as an alternative to pre-defined output distributions (like Gaussians). NFs parametrize non-linear transformations that transform a Gaussian base-distribution into a more expressive density and use that for prediction (Papamakarios et al., 2017).

Examples of (ii) are Monte-Carlo-Dropout, which uses dropout layers in combination with multiple forward passes to approximate samples from the parameter distribution of a Bayesian Neural Network, as well as Langevin Dynamics for Neural Networks and "Stein Variational Gradient Descent" (SVGD), which estimate the parameter distribution via an updated gradient descent algorithm. Both approaches are model agnostic (apart from requiring dropout and gradient descent training).

Table 1: Hyperparameters of all proposed variations of our extension to BALD. While BALD (Gal et al., 2017) was proposed for classificaition and uses categorical entropy, BALD$^{\mathbb{H}}$ uses continuous entropy. A dropout rate of 0.05 was showing the best AL performance across all datasets. A * denotes the optimal dropout rate for each dataset. Optimal dropout rates for each dataset are between 0.008 and 0.05.

| | Param. Sampling | Aggregation | Dist. Function | Drop Train | Drop Eval |
|---|---|---|---|---|---|
| BALD | MC dropout | Shannon Entr. | subtraction | 0.5 | 0.5 |
| NFlows Out | MC dropout | $-\sum \log p$ | subtraction | 0.05 | 0.05 |
| BALD$^{\sigma}$ | MC dropout | std | subtraction | 0.05 | 0.05 |
| BALD$^{\mathbb{H}}$ | MC dropout | Shannon Entr. | subtraction | 0.05 | 0.05 |
| BALSA$^{\text{EMD}}$ | MC dropout | - | EMD | 0.05 | 0.05 |
| BALSA$^{\text{EMD}}_{\text{dual}}$ | MC dropout | - | EMD | * | 0.1 |
| BALSA$^{\text{KL}}$ | MC dropout | - | KL-Div. | 0.05 | 0.05 |
| BALSA$^{\text{KL}}_{\text{dual}}$ | MC dropout | - | KL-Div. | * | 0.1 |

Models from category (i) are (to the best of our knowledge) not capable of distinguishing between aleatoric uncertainty and epistemic uncertainty. However, in Active Learning, we are primarily interested in quantifying the epistemic uncertainty, as this is the only quantity that we can reduce by sampling more data points. For that reason, we chose to extend BALD, a well-known algorithm for AL that uses MC-Dropout. Generally, our proposed method also works for Langevin Dynamics or SVGD, but as they change the training procedure itself by adding new terms and a minimum number of epochs, they are not directly comparable to the bulk of AL algorithms. We compiled an overview of our algorithms in Table 1. Without changing the "Aggregation" or "Distance Function" columns (contents detailed in Section 5) we could replace the parameter sampling with Langevin Dynamics or SVGD. We defer studies of the resulting algorithms to future work.

## 4 RELATED WORK

### DEEP ACTIVE LEARNING FOR REGRESSION

Most approaches for Active Learning for Regression are based on geometric properties of the data, with a few notable approaches of uncertainty sampling that are bound to specific model architectures. Geometric methods include Coreset (Sener and Savarese, 2017), CoreGCN (Caramalau et al., 2021) and TypiClust (Hacohen et al., 2022). All three approaches first embed any candidate point using the current model and apply their distance calculations in latent space. Coreset picks points with maximal distances to each previously sampled point. CoreGCN does one more embedding step by training a Graph Convolutional Model on a node classification task, where each node represents an unlabeled data point. Finally, Coreset sampling is applied in this updated embedding space from the Graph Convolutional Model. TypiClust uses KNN-Clustering to bin the points into $|L^{(i)}| + \tau$ many clusters and then select at most one point from each cluster.

Many UC approaches for AL with regression are not agnostic to the model architecture (Jose et al., 2024; Riis et al., 2022) and cannot directly be applied to our setting with normalizing flows. One of the few exceptions to this is the BALD algorithm itself, as it's only requirement are dropout layers in the model architecture.

### CLOSEST RELATED WORK

The authors of (Berry and Meger, 2023b;a) already researched using normalizing flows in an ensemble and how to approximate this construct via MC dropout. They proposed two different ways of applying dropout masks to normalizing flows: either in the bijective transformations (called NFlows Out) or in a network that parametrizes the base distribution of the normalizing flow (called NFlows Base). Their methods are evaluated on a synthetic uncertainty quantification tasks, as well as a synthetic AL task with random sampling and a fixed query size of $\tau = 10$. We differ from the work of (Berry and Meger, 2023b;a) in the following ways:

Table 2: Characteristics of used datasets for this work. Datasets are selected to cover a large range of size and complexity and provide maximal intersection with other literature for AL with regression

| Name | #Feat | #Inst (Train) | $L^{(0)}$ | $B$ |
|---|---|---|---|---|
| Parkinsons (Tsanas and Little, 2009) | 61 | 3760 | 200 | 800 |
| Supercond. (Hamidieh, 2018) | 81 | 13608 | 200 | 800 |
| Sarcos (Fischer, 2022) | 21 | 28470 | 200 | 1200 |
| Diamonds (Mueller, 2019) | 26 | 34522 | 200 | 1200 |

(i) While (Berry and Meger, 2023b;a) proposes to implement the uncertainty function $\mathbb{H}$ in BALD as $-\sum \log [\hat{y}_\theta(x)]$, we use Shannon-Entropy and propose multiple additional implementations.
(ii) (Berry and Meger, 2023b;a) conducted their experiments solely on synthetic data from simulations and compared NFlows only against other dropout-based AL algorithms. We extend this use case to 4 real world datasets with multiple acquisition functions and query sizes.
(iii) Finally, we opt for applying dropout masks only to the conditioning model and to sample random dropout masks instead of using the fixed masks from (Berry and Meger, 2023b;a). Even though we acknowledge the potential usefulness of these approaches, none of them have yet been tested on pool-based AL on real world data. We focus first on the most natural application of MC dropout for normalizing flows and defer the other versions to future work.

### MONTE-CARLO DROPOUT FOR ACTIVE LEARNING

MC dropout for AL was first proposed by BALD (Gal et al., 2017) as a way to estimate parameter uncertainty (epistemic uncertainty). The core idea of BALD is to sample a model's parameter distribution $p(\theta)$ multiple times and measure the total (aleatoric+epistemic) uncertainty of each sample. As an approximation of aleatoric uncertainty, the authors then measure the uncertainty of the average prediction and contrast that from the uncertainty of each parameter sample to obtain the epistemic uncertainty (Eq. 1). The authors derived their algorithm for softmax-classification with neural networks, but the general idea of measuring the uncertainty of $k$ parameter samples contrasted by the uncertainty of the average prediction is applicable to regression as well.

$$\text{BALD}(x) = \sum_{i=1}^{k} \left( \mathbb{H}\left[\bar{y}(x)\right] - \mathbb{H}\left[\hat{y}_{\theta_i}(x)\right] \right) \tag{1}$$

$$\bar{y}(x) = \frac{1}{k} \sum_{j=1}^{k} \hat{y}_{\theta_j}(x)$$

Natural choices for the uncertainty function $\mathbb{H}$ for predictive distributions are the standard deviation or the Shannon-Entropy. The subtraction in Eq. 1 serves as a distance measure between the total uncertainty of a parameter sample and the uncertainty of the average prediction. Following that idea, if a metric $\phi$ exists that can measure the distance between $\hat{y}_{\theta_i}$ and $\bar{y}$ directly, we can apply the following variant of Eq. 1:

$$\text{BALD}(x) = \sum_{i=1}^{k} \phi\left(\hat{y}_{\theta_i}(x), \ \bar{y}(x)\right) \tag{2}$$

Based on Eq. 2, we are proposing two variants of a novel algorithm, which we call BALSA.

## 5 METHODOLOGY

### BALSA

We define the conditional predictive distribution that a model produces after a point $x$ was fed to the encoder $\psi_\theta$, which conditions the distribution, as $\hat{p}|\psi_\theta(x)$ or $\hat{p}_\theta|x$ for short.
To employ Eq. 2, we have to solve one main problem: how is the "average" predictive distribution $\bar{p}|x$ (analogous to $\bar{y}(x)$) defined? We are proposing two solutions:

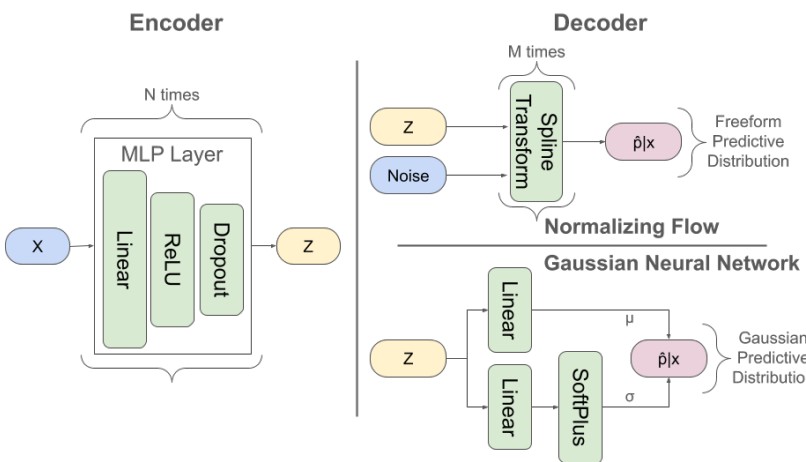

Figure 1: Overview of our regression models. Both models use an MLP encoder to create a latent embedding $z$ of the input, before using $z$ to parametrize a predictive distribution.

**Grid Sampling**   Since there exist no sound way of averaging iid samples (and their likelihoods) from arbitrary distributions to obtain $\bar{p}|x$, we are changing the sampling method to a more rigid structure. To this end we are normalizing our target values between $[0..1]$ during pre-processing and distribute samples on a grid with a resolution of 200. We use our constructed samples to obtain likelihoods from our model and denote the vector of likelihoods on the grid as $\hat{p}_\theta^\rightarrow|x \in \mathbb{R}^{200}$. Finally, we can average multiple likelihood vectors like this across $k$ parameter samples to obtain $\bar{p}|x \in \mathbb{R}^{200}$.

$$\bar{p}|x = \frac{1}{k}\sum_{j=1}^{k}\hat{p}_{\theta_j}^\rightarrow|x$$

As a vector of averaged likelihoods is no longer normalized, we need to re-normalize the values by the area under the curve to obtain a proper distribution. However, we observed in our experiments that the un-normalized version of $BALSA$ performs comparable to or worse than the re-normalized one (We provide the respective ablation study in Sec 7). Therefore, we focus on the un-normalized version and omit the normalization step in our formulas. The formulas including the normalization step can be found in Appendix B.

**Pair Comparison**   To avoid the computation of $\bar{p}|x$ entirely, we propose to approximate Eq. 2 with pairs of parameter samples instead. Given $k$ parameter samples, we define $k-1$ pairs of predictive distributions and measure their distances.

$$\sum_{i=1}^{k-1}\phi\left(\hat{p}_{\theta_i}|x,\ \hat{p}_{\theta_{i+1}}|x\right)$$

Since the parameter samples $\theta_i$ are drawn iid, the sum is not influenced by sequence effects from the ordering of the $k$ samples.

Finally, we need a distance metric $\phi$ to measure the difference between two arbitrary predictive distributions. We propose KL-Divergence and Earth Mover's Distance (EMD) and call our resulting algorithms $BALSA^{\text{KL}}$ and $BALSA^{\text{EMD}}$ respectively.

$BALSA^{\textbf{KL}}$   KL-Divergence is measured on likelihood vectors of two distributions and is proportional to the expected surprise when one distribution is used as a model to describe the other. The higher the surprise, the more different the two distributions are.
Implementing both above mentioned approaches we have a grid sampling version $BALSA^{\text{KL Grid}}$

and a pair comparison version $BALSA^{\text{KL Pair}}$.

$$BALSA^{\text{KL Grid}}(x) = \sum_{i=1}^{k} \text{KL}\left(\hat{p}_{\theta_i}^{\dashv}|x, \ \bar{p}|x\right) \tag{3}$$

$$BALSA^{\text{KL Pair}}(x) = \sum_{i=1}^{k-1} \text{KL}\left(\hat{p}_{\theta_i}|x, \ \hat{p}_{\theta_{i+1}}|x\right) \tag{4}$$

A mathematical analysis of the differences between the resulting $BALSA^{\text{KL Grid}}$ algorithm and BALD can be found in Appendix A. We omitted this analysis for $BALSA^{\text{KL Pair}}$ and the following $BALSA^{\text{EMD}}$, because both use fundamentally different computations and are therefore considered different algorithms.

$BALSA^{\textbf{EMD}}$  The Earth Mover's Distance (a.k.a. Wasserstein Distance) is computed over iid samples drawn from distributions and is proportional to the cost of transforming one distribution into the other. Since EMD relies on iid samples, we cannot use $\hat{p}_{\theta_i}^{\dashv}|x$ in this context. We only implement the pair comparison version, simply called $BALSA^{\text{EMD}}$.

$$BALSA^{\text{EMD}}(x) = \sum_{i=1}^{k-1} \text{EMD}\left(y'_{\theta_i}, \ y'_{\theta_{i+1}}\right) \tag{5}$$

$$y'_{\theta} \sim \hat{p}_{\theta}|x$$

BASELINES

We are using Coreset (Sener and Savarese, 2017), CoreGCN (Caramalau et al., 2021) and Typi-Clust (Hacohen et al., 2022) as clustering based competitors to our uncertainty based algorithms. Additionally, we adapt 3 well-known uncertainty sampling heuristics to models with predictive distributions. Neither the clustering approaches, nor the heuristics rely on MC dropout, hence we omit the index on the parameters $\theta$.

For the heuristics, we measure (i) the standard deviation $\sigma$ of samples from the predictive distribution, (ii) the log likelihood of the most probable prediction (least confident sampling) and (iii) the Shannon entropy of the predictive distribution.

We denote baseline (i) as $Std = \sigma(y'_{\theta})$, which is computed based on 200 samples from the predictive distribution.

We denote baseline (ii) as $LC = -\text{argmax}_{y'} \hat{p}_{\theta}|x(y')$ where the most probable sample is again found by sampling 200 points.

We denote baseline (iii) as $Entr = -\hat{p}_{\theta}|x \log[\hat{p}_{\theta}|x]$. Since we are dealing with regression problems and predictive distributions, we use continuous entropy in this work. Calculating continuous entropy entails integrating $\int -\hat{p}_{\theta}|x \log[\hat{p}_{\theta}|x]\,dx$, which we approximate by employing our grid sampling approach, computing the entropy of the resulting likelihood vector $\hat{p}^{\dashv}|x$ and finding the total entropy with the trapezoidal rule

$$Entr(x) = \text{trapz}\left(-\hat{p}_{\theta}^{\dashv}|x \ \log[\hat{p}_{\theta}^{\dashv}|x]\right) \tag{6}$$

As all baselines (i - iii) are viable replacements of the function $\mathbb{H}$ in BALD (Eq. 1), we can construct additional baselines in a straightforward fashion by creating adaptations of BALD for models with predictive distributions.

Based on baseline (i), we construct $BALD^{\sigma}$. Since the standard deviation needs to be computed over iid samples from $\hat{p}_{\theta}|x$ we use pair comparisons (analogous to $BALSA^{\text{EMD}}$).

$$BALD^{\sigma}(x) = \sum_{i=1}^{k-1}\left(\sigma\left[y'_{\theta_i}\right] - \sigma[y'_{\theta_{i+1}}]\right) \tag{7}$$

$$y'_{\theta} \sim \hat{p}_{\theta}|x$$

Based on baseline (ii), we construct $BALD^{\text{LC}}$. Following $BALD^{\sigma}$, this baseline is also computed over pairs.

$$BALD^{\text{LC}}(x) = \sum_{i=1}^{k-1}\left(LC\left[\hat{p}_{\theta_i}|x\right] - LC[\hat{p}_{\theta_{i+1}}|x]\right) \tag{8}$$

Based on baseline (iii), we construct $BALD^{\mathbb{H}}$. To stay as close as possible to BALD, $BALD^{\mathbb{H}}$ uses $\hat{p}_\theta^{\rightarrow}|x$ to compute $\bar{p}|x$ and reproduces Eq. 1.

$$BALD^{\mathbb{H}}(x) = \sum_{i=1}^{k} \left( Entr\left[\bar{p}|x\right] - Entr\left[\hat{p}_\theta^{\rightarrow}|x\right] \right) \tag{9}$$

$$\bar{p}|x = \frac{1}{k} \sum_{j=1}^{k} \hat{p}_{\theta_j}^{\rightarrow}|x$$

## 6    IMPLEMENTATION DETAILS

All experiments are run with PyTorch on Nvidia 2080, 3090 and 4090 GPUs. The total runtime for all experiments was approximately 7 days on 40-50 GPUs.

As backbone model we are using a standard MLP encoder with dropout layers and ReLU activation. The encoder is conditioning the predictive distribution of our model either via a $\mu$-decoder and a $\sigma$-decoder (GNN) or as a conditioning input for the normalizing flow. Our normalizing flow is an autoregressive Neural Spline Flow with rational-quadratic spline transformations (Durkan et al., 2019). For detailed descriptions on both models, please refer to Appendix C. We optimize all our hyperparameters on random subsets of size $B$ (e.g. Parkinsons has a budget of 800). To that end, we evaluate any hyperparameter setting on 4 different random subsets and use average validation performance as metric for our search.

Evaluating algorithms that include MC dropout is especially tricky, as few guidelines exist on how to choose an appropriate dropout rate. Instead of forcing a (too) high dropout rate onto every algorithm, in this work we include dropout in our hyperparameter search so it will be optimized for validation performance on each dataset. This creates an optimal evaluation scheme for algorithms without MC dropout. This is an important step in order to not underestimate the performance of algorithms that do not require high dropout rates. We then let each BALD or BALSA algorithm overwrite the dropout rate to a fixed value. The specific rate of MC dropout for overwriting the default setting is optimized for AL performance across all datasets on very few trials in order to find a suitable default value. Finally, we propose an alternative to overwriting the optimal dropout rate to a fixed value: We test BALSA in "dual" mode, retaining the optimal dropout during training and switching to a higher fixed value during evaluation phases. A fixed evaluation rate of 0.05 is chosen as the highest of our optimal dropout rates (0.008-0.05 per dataset). This is still a full magnitude lower than common rates of 0.5 for MC dropout in the literature (Gal et al., 2017; Kirsch et al., 2019). Please refer to Table 1 for the dropout settings of our algorithms and Appendix C for our used hyperparameters. The results for "dual" mode can be found in Section 7 in the respective ablation study.

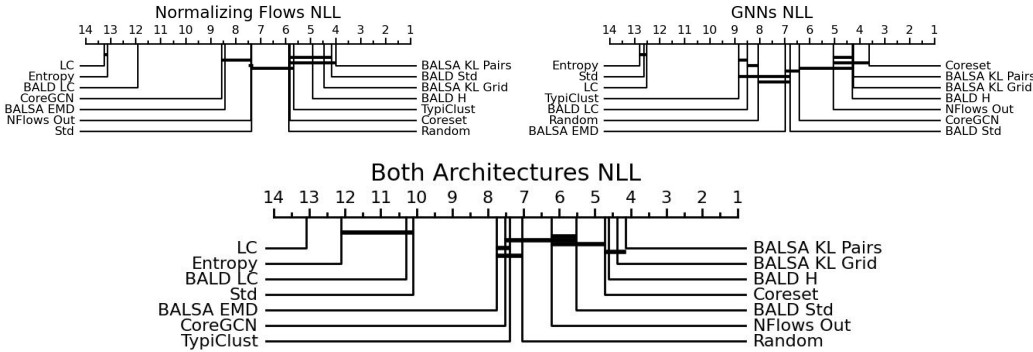

Figure 2: Critical Difference Diagram for all datasets and query size 1. (lower is better) Horizontal bars indicate statistical significance according to the Wilcoxon-Holm test.

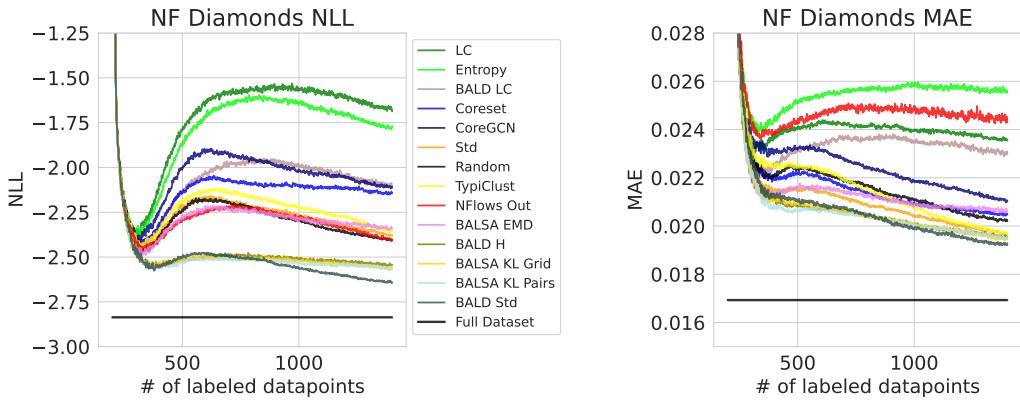

Figure 3: AL trajectories of all tested algorithms in the Diamonds dataset. Curves based on NLL (left) and MAE (right); lower is better. Trajectories are averaged over 30 restarts of each experiment.

# 7 EXPERIMENTS

We test our proposed algorithms from Section 5 on conditional normalizing flows and GNNs on 4 datasets (Details of our datasets in Table 2) and across query sizes of $\tau = \{1, 50, 200\}$. Every experiment is repeated 30 times and implemented according to the guidelines of (Ji et al., 2023) and (Werner et al., 2024). We compare our results mainly on query size 1, as we are mostly interested in the ability of our proposed algorithms to capture uncertainty in the model rather than adapting to larger query sizes. Following (Werner et al., 2024) we choose CD-Diagrams as aggregation method for comparison. To this end, we compute a ranking of each algorithm's AUC value for each dataset and for each repetition and compare the ranks via the Wilcoxon signed-rank test. Computing ranks out of the AUC values enables us to compare results across datasets without averaging AUC values from different datasets. The AUC values are computed based on test NLL (Fig. 2) and test MAE (Fig. 4). For context, we evaluate Coreset (Sener and Savarese, 2017), CoreGCN (Caramalau et al., 2021) and TypiClust (Hacohen et al., 2022) and display the final ranking across all datasets on query size 1 in Figure 2. Additionally, we exemplarily display the AL trajectories of all algorithms for the Diamonds dataset in Figure 3. The remaining figures for all datasets can be found in Appendix D. In our experiments, $BALSA^{\text{KL Pairs}}$ is the best AL algorithm on average, followed by $BALSA^{\text{KL Grid}}$, $BALD^{\mathbb{H}}$ and Coreset. Notably, common AL heuristics, namely the Shannon Entropy, Std and Least Confidence baselines, which usually are among the most reliable methods for AL with classification, performed especially bad. These results indicate, that not every kind of measure on the uncertainty quantification is useful for AL, even when the UC is inert to the model architecture and the measure is well-tested in other domains. Interestingly, Coreset and CoreGCN perform a lot better with GNN architectures, both gaining about 3 ranks, while TypiClust - the also a clustering algorithm - loses ranks. To investigate and compare these algorithms further, we provide additional results in Figure 4, computing the ranks of each algorithm based on MAE instead of NLL. The two main differences are (i) Nflows Out loses drastically, scoring last on average and (ii) Coreset is now the best performing algorithm, winning closely against $BALSA^{\text{KL Pairs}}$ and TypiClust.

Finding the right (mix of) metrics to evaluate our models remains a challenging task, as every chosen measure inevitably introduces a bias into the evaluation. Since we have optimized our hyperparameters for validation NLL, we opt for NLL as our main metric. We have included results for our main experiments (Fig. 2) measured with the CRPS score instead of NLL in Appendix E. The CRPS score resulted in the same ranking as likelihood did, so we opted to use the less involved score.

Additionally, we provide multiple ablation studies for our proposed BALSA algorithm:

**Dual Mode:** We test BALSA in "dual" mode by switching between the optimal dropout and a static value during training and evaluation phases respectively. This approach poses an alternative to the highlighted problems of setting dropout rates described in Section 6. Unfortunately, the results in Figure 5 are inconclusive, as across all datasets and model architectures the dual mode archives one clear loss ($BALSA_{\text{dual}}^{\text{EMD}}$), a marginal loss ($BALSA_{\text{dual}}^{\text{KL Pairs}}$) and a marginal win ($BALSA_{\text{dual}}^{\text{KL Grid}}$). We hypothesize that the switch of dropout rate between training and evaluation can in some cases

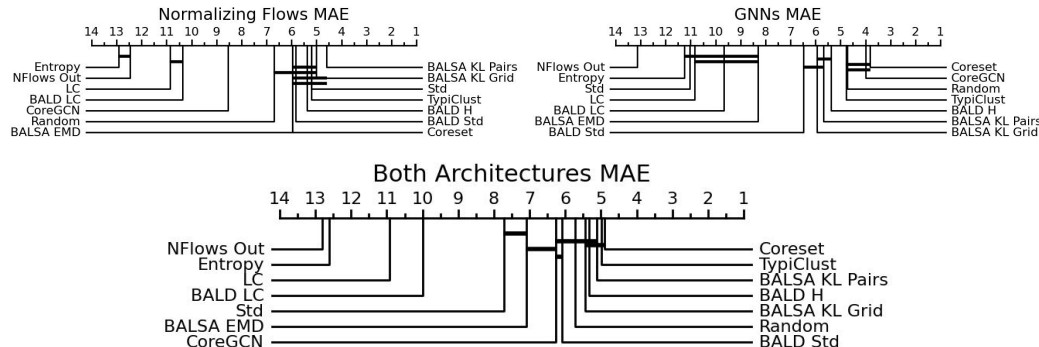

Figure 4: Critical Difference Diagrams with ranks computed based on MAE instead of NLL. Same experimental parameters as Fig. 2

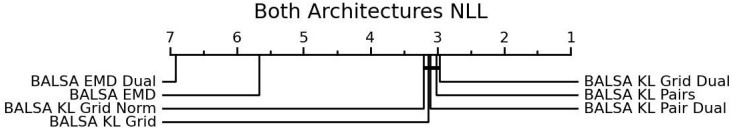

Figure 5: Comparison of "dual" evaluation mode for both BALSA algorithms as well as the re-normalized version of $BALSA^{\text{KL Grid}}$. Based on NLL and $\tau = 1$

degrade the models prediction to much, as the model was not trained to cope with higher than optimal dropout.

**Re-normalization:** We also tested a version of $BALSA^{\text{KL Grid}}$ where we re-normalize $\bar{p}|x$ with its area under the curve as described in Section 5. We included $BALSA^{\text{KL Grid}}_{\text{norm}}$ in Figure 5, but observed the slightly lower performance compared to un-normalized $BALSA^{\text{KL Grid}}$. For the sake of brevity and simplicity, we therefore opt to leave the normalization step out of our formulas.

**Query Sizes:** To gauge how well our proposed variants of BALD and BALSA adapt to larger query sizes, we test our proposed methods on $\tau = \{50, 200\}$ and compare the results in Figure 6. For ease of comparison, we exclude the 4 worst performing algorithms. Interestingly, when increasing the query size $\tau$, clustering algorithms like Coreset and TypiClust are losing performance more quickly than our proposed uncertainty sampling methods. This finding contradicts experiments on AL for classification, where those methods are very stable as $\tau$ increases (Ji et al., 2023; Werner et al., 2024). The uncertainty sampling methods are behaving as expected, gradually losing their advantage over random sampling with increasing query size, as they suffer from missing diversity sampling components.

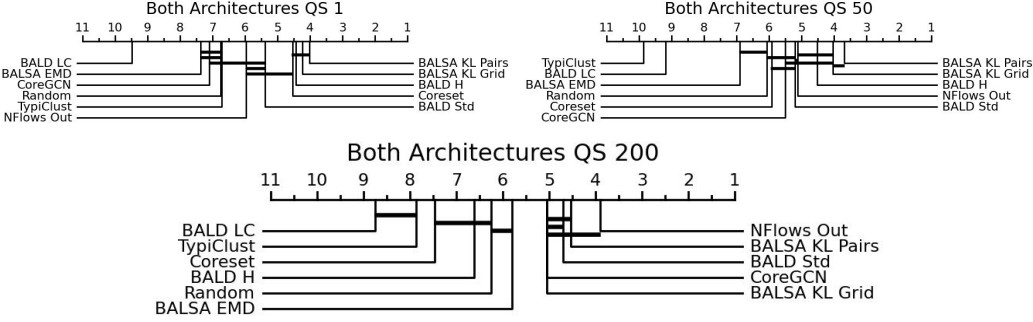

Figure 6: Comparison of our best performing algorithms across different query sizes. Both model architectures, based on NLL.

## 8 CONCLUSION

In this work, we extended the foundation of (Berry and Meger, 2023b;a) by applying the idea of using MC dropout normalizing flows to real world data and pool-based AL. To that end, we adapted 3 heuristic AL baselines to models with predictive distributions, proposed 3 straightforward adaptations of BALD and created 2 novel algorithms, based on the BALD algorithm. This creates a comprehensive benchmark suite for uncertainty sampling for the use case of AL with models with predictive distributions. We demonstrate strong performance across 4 datasets for normalizing flows for $BALSA^{\text{KL Pairs}}$, narrowly losing against Coreset for GNN models. For larger query sizes, we observed unexpected behavior for clustering algorithms like Coreset and TypiClust, which were falling behind uncertainty based algorithms for $\tau = \{50, 200\}$, while uncertainty based algorithms retain their performance. This goes against common knowledge in AL, which attributes high potential to clustering algorithms to scale to larger query sizes. This work is but the first step to understanding the dynamics of AL for regression models with uncertainty quantification.

## REPRODUCIBILITY STATEMENT

Our code is publicly available under: `https://anonymous.4open.science/r/Bayesian-Active-Learning-By-Distribution-Disagreement-8682/`
We did not provide pseudo-code or algorithms for our experiments, because our setup is identical to (Werner et al., 2024). Please refer to their work for details.
The employed hyperparameters can be found in Appendix C or in the "configs" folder in the code.

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

# A DIFFERENCE BETWEEN BALD AND $BALSA^{\text{KL}}$

$$\text{BALD}(x \mid \hat{p}_{1:K}) := \sum_{k=1}^{K} \mathbb{H}(\bar{p}(y \mid x)) - \mathbb{H}(\hat{p}_k(y \mid x)), \quad \text{with } \bar{p}(y|x) := \tfrac{1}{K} \sum_{k=1}^{K} \hat{p}_k(y|x)$$

Let $\text{KL}(p, q) := \int_y p(x) \log \frac{p(y)}{q(y)} dy$ be Kullback-Leibler divergence.

$$\text{BALSA}(x \mid \hat{p}_{1:K}) := \sum_{k=1}^{K} \text{KL}(\hat{p}_k(y \mid x)), \bar{p}(y \mid x))$$

To see the differences between bald and balsa more clearly: write the $k$-th balsa term shorter dropping the throughout dependency on $x$:

$$\text{KL}(\hat{p}_k, \bar{p}) = \int \hat{p}_k(y) \log \frac{\hat{p}_k(y)}{\bar{p}(y)} dy$$

$$= \int \hat{p}_k(y) \log \hat{p}_k(y) dy - \int \hat{p}_k(y) \log \bar{p}(y) dy$$

$$= -H(\hat{p}_k(y)) - \int \hat{p}_k(y) \log \bar{p}(y) dy$$

which is different from the $k$-th bald term

$$\text{BALD}(x \mid \hat{p}_k) = \mathbb{H}(\bar{p}(y \mid x)) - \mathbb{H}(\hat{p}_k(y \mid x))$$

$$= -\mathbb{H}(\hat{p}_k(y)) + \int \bar{p}(y) \log \bar{p}(y) dy$$

# B $BALSA^{\text{KL Grid}}$ WITH NORMALIZATION

Formulas for $BALSA^{\text{KL Grid}}_{\text{Norm}}$ as tested in the respected ablation in Section 7.
We found this version to perform identical to the un-normalized version of $BALSA^{\text{KL Grid}}$ and opted for the less involved formulation.

$$BALSA^{\text{KL Grid}}(x) = \sum_{i=1}^{k} \text{KL}\left(\hat{p}_{\theta_i}^{\dashv}|x, \ \frac{\bar{p}|x}{\text{trapz}(\bar{p}|x)}\right)$$

$$\bar{p}|x = \frac{1}{k} \sum_{j=1}^{k} \hat{p}_{\theta_j}^{\dashv}|x$$

$$\text{trapz}(p^{\dashv}) = \sum_{n=1}^{|p^{\dashv}|-1} \frac{1}{2}\left(p_n^{\dashv} + p_{n+1}^{\dashv}\right)$$

The trapz-method is a well-known method to approximate an integral. We use the PyTorch-Implementation of trapz.

# C MODEL ARCHITECTURES

We use a MLP encoder model for both architectures. In our Normalizing Flow models, the encodings are used as conditioning input for the bijective transformations (decoder). Our GNNs use a linear layer to decode $\mu$ and $\sigma$ from the encodings.

Our Normalizing Flow model is a masked autoregressive flow with rational-quadratic spline transformations, which has demonstrated good performance on a variety of tasks in (Durkan et al., 2019).

Table 3: Used Hyperparameters for Normalizing Flow models

|  | Parkinsons | Diamonds | Supercond. | Sarcos |
|---|---|---|---|---|
| Encoder | [32, 64, 128] | [32, 64, 128] | [32, 64, 128] | [32, 64, 128] |
| Decoder | [128, 128] | [128, 128] | [128, 128] | [128, 128] |
| Budget | 800 | 1200 | 800 | 1200 |
| Seed Set | 200 | 200 | 200 | 200 |
| Batch Size | 64 | 64 | 64 | 64 |
| Optimizer | NAdam | NAdam | NAdam | NAdam |
| LR | 0.001 | 0.0004 | 0.0008 | 0.0007 |
| Weight Dec. | 0.0018 | 0.008 | 0.0003 | 0.0004 |
| Dropout | 0.0163 | 0.0194 | 0.0491 | 0.0261 |

Table 4: Used Hyperparameters for GNN models

|  | Parkinsons | Diamonds | Supercond. | Sarcos |
|---|---|---|---|---|
| Encoder | [32, 64, 128] | [32, 64, 128] | [32, 64, 128] | [32, 64, 128] |
| Decoder | linear | linear | linear | linear |
| Budget | 800 | 1200 | 800 | 1200 |
| Seed Set | 200 | 200 | 200 | 200 |
| Batch Size | 64 | 64 | 64 | 64 |
| Optimizer | NAdam | NAdam | NAdam | NAdam |
| LR | 0.0007 | 0.0004 | 0.0003 | 0.0006 |
| Weight Dec. | 0.0008 | 0.005 | 0.005 | 0.0009 |
| Dropout | 0.0077 | 0.0122 | 0.0121 | 0.0074 |

# D    AL TRAJECTORIES

## PAKINSONS

### Normalizing Flows

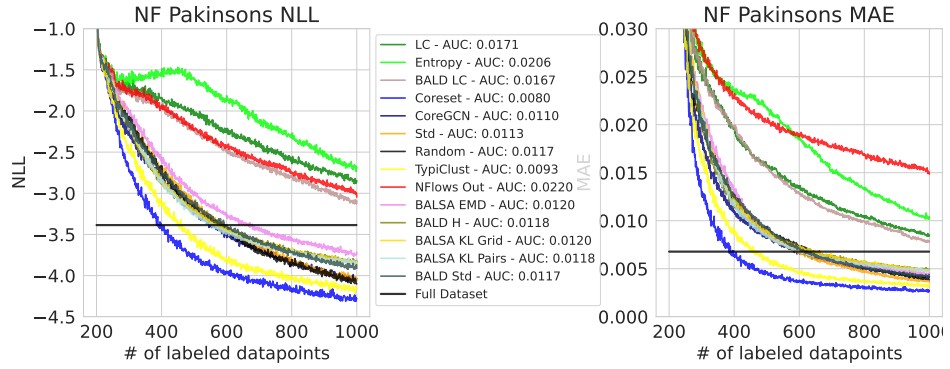

### Gaussian Neural Networks

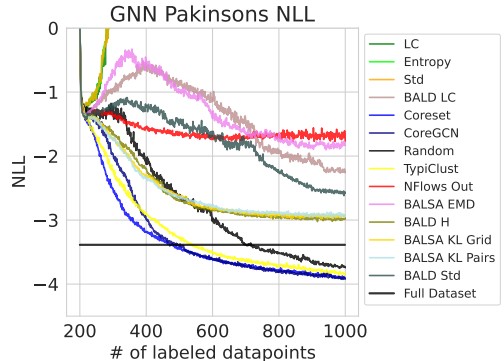
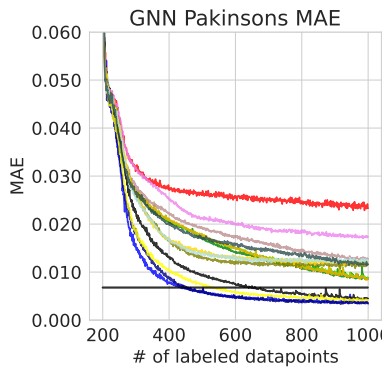

## DIAMONDS

### Normalizing Flows

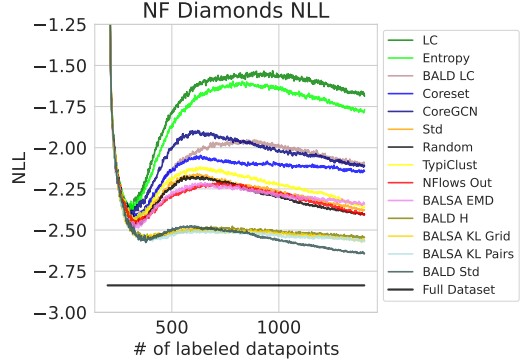
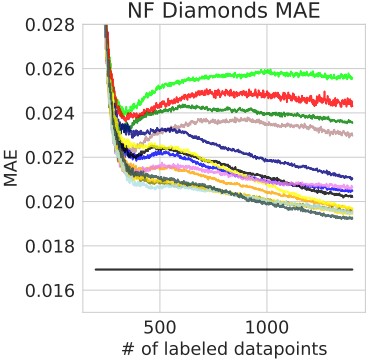

**Gaussian Neural Networks**

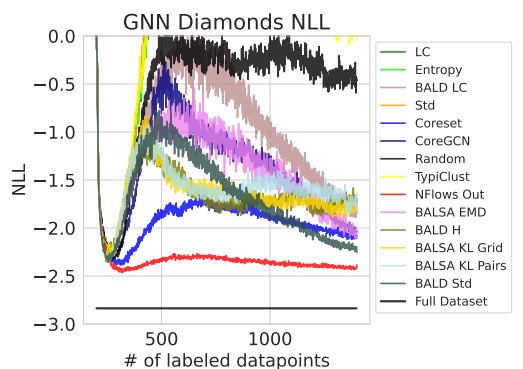
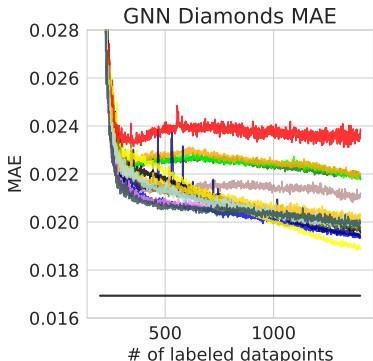

SARCOS

**Normalizing Flows**

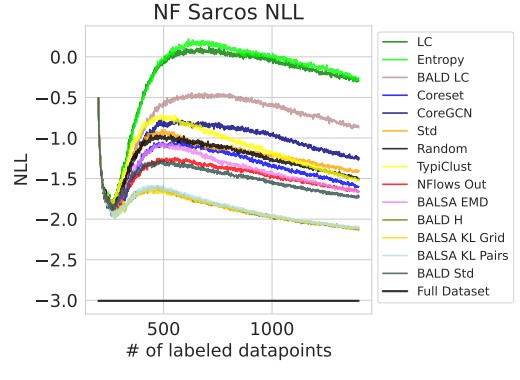
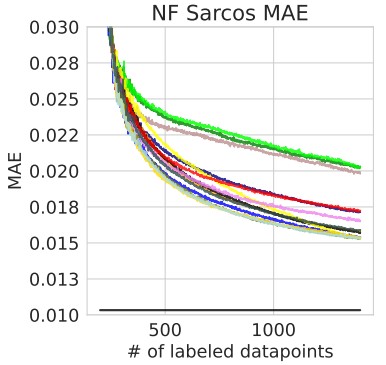

**Gaussian Neural Networks**

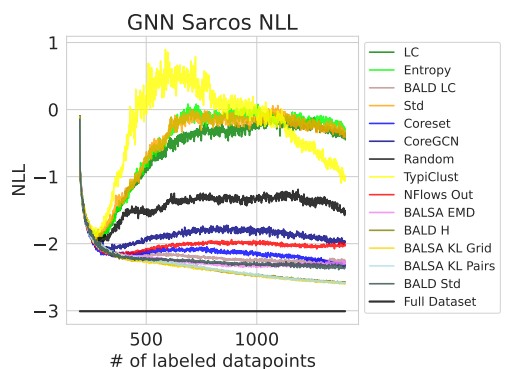
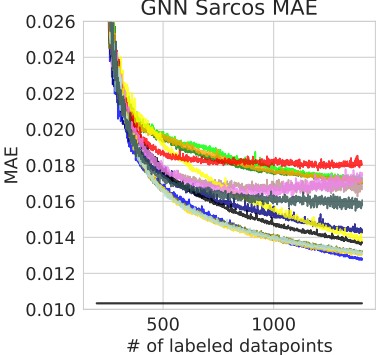

SUPERCONDUCTORS

**Normalizing Flows**

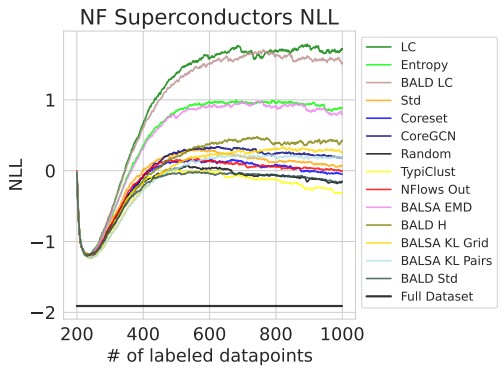

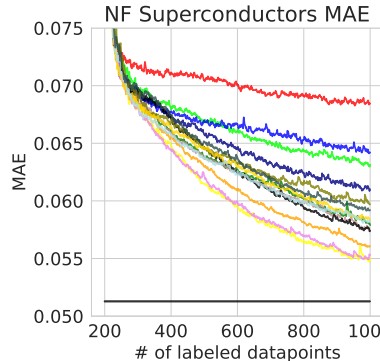

**Gaussian Neural Networks**

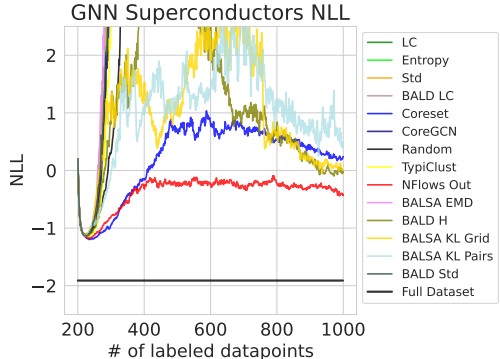

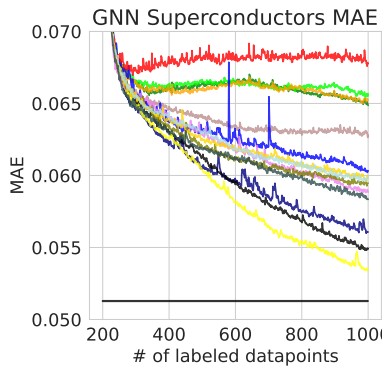

# E  CRPS RESULTS

Reproduction of our main experiment (Figure 2) with ranks computed based on CRPS score instead of NLL. The ranking is identical to Fig. 2, so we opted for the less involved metric for our experiments.

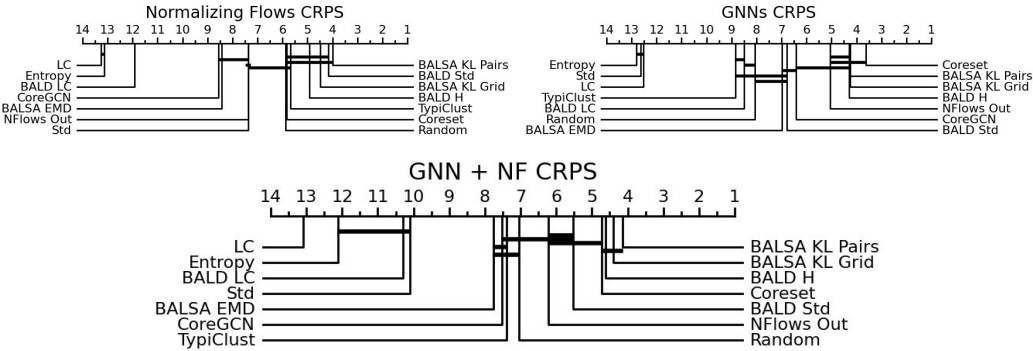