# OpenReview forum: "Bayesian Active Learning By Distribution Disagreement"
_ICLR.cc/2025/Conference — Submitted to ICLR 2025_

### Official Review · Reviewer_pif1 · 2024-10-23

**Soundness:** 2
**Presentation:** 1
**Contribution:** 1
**Rating:** 3
**Confidence:** 3

**Summary:**

The paper focuses on active learning for regression, studying two model classes: normalizing flows and Gaussian Neural Networks (GNNs). Authors show that standard active learning acquisition functions (like entropy minimization) do not work well when applied to these models. Instead, authors apply approximate Bayesian inference in form of MC-dropout, and propose several adaptations of the previously proposed BALD acquisition function. Authors evaluate the baselines and proposed methods on 4 regression datasets.

**Strengths:**

- Originality: while I am not an expert in active learning, to the best of my knowledge the grid and pair-wise versions of the BALD acquisition function are novel.
- Reproducibility: comprehensive experimental details provided in the paper, code is publicly available.
- Authors use critical difference diagrams to report results, which allows to compare methods across all datasets at once.

**Weaknesses:**

- Lack of focus and motivation: the paper in its current form lacks focus, and does not present a compelling story. It proposes several methods and baselines, but the narrative did not make it clear what problems (with existing methods) the proposed methods are supposed to solve. The paper makes several observations, but the overall message/recommendation of the paper is not clear to me.
- Clarity: it took me multiple re-reads to fully grasp the proposed methods and baselines. For example, Eq. (2) assumes a deterministic label, while section 5 suddenly jumps to predictive distributions. The "baselines" part of Section 5 is very dense. Section 3 does not cite important work (e.g. SVGD). Figure 1 is never mentioned in the main text. Finally, authors have tweaked the style to remove paragraph spacing (presumably to fit more content), which further degrades readability.
- Significance: results suggest that BALD H and BALD Std are more or less competitive with BALSA KL variants, both on NLL and MAE. It is not clear to me from the results that BALSA strongly outperforms BALD. Combined with the lack of clear motivation for the method, this does not convince me that I should choose BALSA over BALD. The observation that baselines that don't estimate epistemic uncertainty don't do well in active learning is not surprising to me: I believe it's known that epistemic uncertainty is much more important for active learning. While the paper prods in multiple interesting directions in active learning, I did not find a significant-enough contribution to recommend acceptance.

**Questions:**

- If authors were to pick the most significant/surprising finding in the paper, what would they pick?
- Why are authors only comparing _subsequent_ pairs in pairwise methods? Have authors considered performing comparisons across all pairs of posterior samples? Could this improve performance? (Appreciating this might have an impact on computational complexity)
- Is the 200 grid size fixed across all experiments? Have authors tried to perturb this value, and does it have any measurable impact on performance?
- What are the computational complexities (or empirical runtimes) of the evaluated methods/baselines?

---

> ### Author Response · Authors · 2024-11-18
> **Rebuttal**
>
> Dear reviewer, \
> we thank you for your effort and constructive feedback.  \
> We will incorporate your points into our work in the future, but since this will take considerable time we have elected not to submit a rebuttal for this round.
>
> Best Regards\
> The Authors

---

### Official Review · Reviewer_g8ps · 2024-10-23

**Soundness:** 2
**Presentation:** 3
**Contribution:** 2
**Rating:** 3
**Confidence:** 4

**Summary:**

The authors developed an active learning (AL) algorithm for regression, introducing BALSA, an adaptation of the BALD algorithm specifically designed for regression tasks. They extended the Shannon Entropy used in BALD by incorporating additional distance measures, including KL-Divergence and Earth Mover’s Distance (EMD). The proposed method was evaluated on four datasets.

**Strengths:**

The authors gave a clear explanation of their proposed methods.

**Weaknesses:**

(1) The baseline models used in this paper are limited. Several active learning models capable of handling regression problems, such as BADGE [1], SIMILAR [2], and BAIT [3], are not included or compared in the experiments.

(2) Many other Bayesian models are not considered. The use of Bayesian neural networks (BNNs) is restricted, with key methods such as SGHMC [4], SG-MCMC [5], and cSG-MCMC [6] missing.

(3) Regarding data modality, only tabular data are included, while image-based datasets, such as IMDB-WIKI [7], could have been explored.

(4) There is also limited discussion and experimentation on key factors such as query batch size and the differences between low-data and high-data domains.

(5) No theoretical analysis or insights are provided to explain the performance improvements or offer any guarantees.

Minor suggestions:

(1) AL for regression should be reflected in the title.

(2) Methods parts are too long for this 9-page paper.


References:

[1] Ash, Jordan T., Chicheng Zhang, Akshay Krishnamurthy, John Langford, and Alekh Agarwal. "Deep batch active learning by diverse, uncertain gradient lower bounds." arXiv preprint arXiv:1906.03671 (2019).

[2] Kothawade, Suraj, Nathan Beck, Krishnateja Killamsetty, and Rishabh Iyer. "Similar: Submodular information measures based active learning in realistic scenarios." Advances in Neural Information Processing Systems 34 (2021): 18685-18697.

[3] Ash, Jordan, Surbhi Goel, Akshay Krishnamurthy, and Sham Kakade. "Gone fishing: Neural active learning with fisher embeddings." Advances in Neural Information Processing Systems 34 (2021): 8927-8939.

[4] Chen, Tianqi, Emily Fox, and Carlos Guestrin. "Stochastic gradient hamiltonian monte carlo." In International conference on machine learning, pp. 1683-1691. PMLR, 2014.

[5] Welling, Max, and Yee W. Teh. "Bayesian learning via stochastic gradient Langevin dynamics." In Proceedings of the 28th international conference on machine learning (ICML-11), pp. 681-688. 2011.

[6] Zhang, Ruqi, Chunyuan Li, Jianyi Zhang, Changyou Chen, and Andrew Gordon Wilson. "Cyclical stochastic gradient MCMC for Bayesian deep learning." arXiv preprint arXiv:1902.03932 (2019).

[7] Rothe, Rasmus, Radu Timofte, and Luc Van Gool. "Deep expectation of real and apparent age from a single image without facial landmarks." International Journal of Computer Vision 126, no. 2 (2018): 144-157.

**Questions:**

Various strategies can be employed to effectively select a batch of queries, such as greedy selection in BatchBALD [1] and stochastic batch acquisition [2]. Could the authors clarify why these extensions to a batch setting were not explored?

References:

[1] Kirsch, Andreas, Joost Van Amersfoort, and Yarin Gal. "Batchbald: Efficient and diverse batch acquisition for deep bayesian active learning." Advances in neural information processing systems 32 (2019).

[2] Kirsch, Andreas, Sebastian Farquhar, Parmida Atighehchian, Andrew Jesson, Frederic Branchaud-Charron, and Yarin Gal. "Stochastic batch acquisition: A simple baseline for deep active learning." arXiv preprint arXiv:2106.12059 (2021).

---

> ### Author Response · Authors · 2024-11-18
> **Rebuttal**
>
> Dear reviewer, \
> we thank you for your effort and constructive feedback.  \
> We will incorporate your points into our work in the future, but since this will take considerable time we have elected not to submit a rebuttal for this round.
>
> Best Regards\
> The Authors

---

### Official Review · Reviewer_8VoG · 2024-10-30

**Soundness:** 1
**Presentation:** 2
**Contribution:** 2
**Rating:** 3
**Confidence:** 5

**Summary:**

The paper proposes BALSA, a novel AL method that addresses challenges in regression models with uncertainty quantification, specifically for pool-based AL. While traditional uncertainty measures like entropy and least confidence are less effective here, BALSA adapts the BALD algorithm to handle predictive distributions, distinguishing between aleatoric and epistemic uncertainty. BALSA demonstrates state-of-the-art results across four datasets and two architectures by leveraging Bayesian techniques like MC Dropout.

**Strengths:**

1. The paper addresses an under-researched area of the literature.
2. The broader approach to the BALD acquisition criterion is interesting and appears to have strong potential.

**Weaknesses:**

1. Formatting issues:
   - Ensure paragraphs are indented throughout the document.
   - Correct the initial quotation marks by using `` for opening quotes in LaTeX.
   - In Figure 3, the abundance of methods makes the lines difficult to differentiate. Consider moving some of these methods to the appendix to enhance readability.

2. Results presentation: The results lack standard deviation and confidence intervals, making it challenging to fully trust the conclusions. Including these would provide clearer insight into the variability and reliability of the findings.

3. Missing baselines and experimental settings: Essential settings from Berry and Meger (2023a, b), such as the 1D and multi-D configurations, are absent, as well as Nflows Base and PaiDEs. These settings serve as crucial baselines and should be included to enable comprehensive comparisons.

4. Terminology clarification: The term MC dropout is used incorrectly in the context of Nflows Out, which is trained using a fixed set of dropout masks rather than MC dropout. This distinction should be clearly communicated to avoid confusion.

5. Acquisition function clarification: Berry and Meger (2023a, b) use differential entropy in their derivation of BALD as the acquisition function. What is meant by line 172 point (i)?

**Questions:**

Is the use of an encoder justified for this problem? It seems potentially excessive; was it essential for achieving strong performance, or could a simpler model have sufficed?

---

> ### Author Response · Authors · 2024-11-18
> **Rebuttal**
>
> Dear reviewer, \
> we thank you for your effort and constructive feedback.  \
> We will incorporate your points into our work in the future, but since this will take considerable time we have elected not to submit a rebuttal for this round.
>
> Best Regards\
> The Authors

---

> > ### Comment · Reviewer_8VoG · 2024-12-02
> >
> > Good luck with your next submission!

---

### Official Review · Reviewer_eLnM · 2024-11-02

**Soundness:** 2
**Presentation:** 1
**Contribution:** 2
**Rating:** 3
**Confidence:** 4

**Summary:**

The paper studies active learning for regression tasks. It recognizes that this is harder than classification tasks and that current methods are suboptimal. The paper proposes 'BALSA' algorithms which are an extension of 'BALD' algorithm. A number of experiments are carried out to suggest the proposed algorithms compare favorably to previous ones.

**Strengths:**

The authors report strong results on the chosen datasets.

**Weaknesses:**

The writing is very poor and hard to follow. The notation exhibits a lack of rigor and mathematical expressions are not properly introduced. For example, it is unclear what each \theta_i in eq. (1) refers to. It's unclear how these quantities are measured.  'BALD' is also mentioned many times and never properly described.

While the authors claim that the datasets were selected to 'provide maximal intersection with other literature for AL with regression', they are quite small by modern standards.

I find the use of 'critical difference diagrams' quite strange, instead of reporting the actual performance of the methods.

I'm not necessarily interested in seeing the code at this stage but the link does not work for me.

**Questions:**

What *exactly* is your algorithm? How do you choose which points to label? Do you compute BASLA(x) for each of the unlabeled points?

How can you tell if your method better captures 'aleatoric' or 'epistemic' uncertainty?

236 - why is there no sound way of doing this?
246 - I don't understand; averaging k distributions still leads to a distribution
260 - how exactly are the 'parameter samples \theta_i' drawn? Why is this pairwise approximation sound?

---

> ### Author Response · Authors · 2024-11-18
> **Rebuttal**
>
> Dear reviewer, \
> we thank you for your effort and constructive feedback.  \
> We will incorporate your points into our work in the future, but since this will take considerable time we have elected not to submit a rebuttal for this round.
>
> Best Regards\
> The Authors

---

### Official Review · Reviewer_2ouR · 2024-11-04

**Soundness:** 2
**Presentation:** 2
**Contribution:** 3
**Rating:** 5
**Confidence:** 2

**Summary:**

The paper addresses the challenges of active learning (AL) research in regression tasks, specifically when using normalizing flow models in pool-based AL settings. The authors propose two novel extensions of BALD algorithm, named BALSA, which approximate the BALD acquisition function by calculating distance between pairs of predictive distributions from different parameter samples. They evaluate BALSA's  performance across four datasets and two model architectures, demonstrating its effectiveness and robustness in real-world applications.

**Strengths:**

The paper is original in its focus on developing active learning strategies specifically for regression with normalizing flows, while much of the AL research traditionally focuses on classification tasks.

The paper offers a comprehensive benchmark for AL in regression with predictive distributions. The experiments are robust, testing BALSA across four diverse regression datasets and two model architectures. This extensive comparison proves BALSA's effectiveness and generalizability.

**Weaknesses:**

The presentation of this paper could be improved.

For example, Figure 1 is presented without reference or explanation in the text, which reduces clarity for the reader.

In Pair Comparison, the paper introduces a pairwise approach to approximate Eq. 2, which is one of the core components of BALSA. However, the paper does not adequately explain why this approach is effective, nor does it discuss any potential trade-offs or advantages that led to this specific choice.

The paper claims that  BALSA is specifically designed for use with Normalizing Flow models, but figure 1 suggests that the method might also be applicable to Gaussian Neural Network. This raises questions about what unique properties of normalizing flows motivated the development of BALSA.  A deeper discussion on this point would clarify the applicability of the method.

**Questions:**

In line 255, did you consider any alternative pairing strategies beyond consecutive pairs, such as randomly selected pairs or stratified pairing? If so, what were the findings or reasons for not using these alternatives?

in lines 278-280,  Why was the mathematical analysis omitted for BALSAKL Pair and BALSAEMD? Given their differences, might an appendix providing an overview of the theoretical differences still be valuable for readers?

In Figure 3, the results suggest potential overfitting, as performance decrease with an increasing number of labeled data points.

---

> ### Author Response · Authors · 2024-11-18
> **Rebuttal**
>
> Dear reviewer, \
> we thank you for your effort and constructive feedback.  \
> We will incorporate your points into our work in the future, but since this will take considerable time we have elected not to submit a rebuttal for this round.
>
> Best Regards\
> The Authors

---

### Meta-Review · Area_Chair_Bkjc · 2024-12-21

**Metareview:**

This paper proposes BALSA, a novel extension of the BALD acquisition function for active learning (AL) in regression tasks using normalizing flows. The method aims to improve the differentiation between aleatoric and epistemic uncertainties, a key challenge in AL for regression. The authors evaluate BALSA on four datasets and two architectures, reporting state-of-the-art performance compared to existing methods.

**Pros**
* The paper addresses an under-researched area of active learning for regression tasks using normalizing flows
* The proposed BALSA method represents a novel extension of the BALD acquisition function

**Cons**
* The paper suffers from clarity and presentation issues, making it difficult to follow.
* Comparisons with relevant baselines (e.g., BADGE, SIMILAR, BAIT) are missing
* Theoretical analysis and insights are lacking
* Empirical results are limited in scope

In summary, while the paper has some interesting contributions, the clarity, completeness of experimental comparisons, and theoretical justification fall short of the standard for acceptance.

**Additional Comments On Reviewer Discussion:**

The authors acknowledged the concerns provided by the reviewers, and proposed to incorporate them into a future revision.

---

### Decision · Program_Chairs · 2025-01-22

Reject